# Computational Analysis of SARS-CoV-2 and SARS-Like Coronavirus Diversity in Human, Bat and Pangolin Populations

**DOI:** 10.3390/v13010049

**Published:** 2020-12-30

**Authors:** Nicholas J. Dimonaco, Mazdak Salavati, Barbara B. Shih

**Affiliations:** 1Institute of Biological, Environmental and Rural Sciences, Aberystwyth University, Wales SY3 3FL, UK; 2The Roslin Institute, Royal (Dick) School of Veterinary Studies, University of Edinburgh, Easter Bush, Midlothian EH25 9RG, UK

**Keywords:** coronavirus, hackathon, host-associated divergences, codon usage, variant discovery

## Abstract

In 2019, a novel coronavirus, SARS-CoV-2/nCoV-19, emerged in Wuhan, China, and has been responsible for the current COVID-19 pandemic. The evolutionary origins of the virus remain elusive and understanding its complex mutational signatures could guide vaccine design and development. As part of the international “CoronaHack” in April 2020, we employed a collection of contemporary methodologies to compare the genomic sequences of coronaviruses isolated from human (SARS-CoV-2; n = 163), bat (bat-CoV; n = 215) and pangolin (pangolin-CoV; n = 7) available in public repositories. We have also noted the pangolin-CoV isolate MP789 to bare stronger resemblance to SARS-CoV-2 than other pangolin-CoV. Following de novo gene annotation prediction, analyses of gene–gene similarity network, codon usage bias and variant discovery were undertaken. Strong host-associated divergences were noted in ORF3a, ORF6, ORF7a, ORF8 and S, and in codon usage bias profiles. Last, we have characterised several high impact variants (in-frame insertion/deletion or stop gain) in bat-CoV and pangolin-CoV populations, some of which are found in the same amino acid position and may be highlighting loci of potential functional relevance.

## 1. Background

The continued and increasing occurrence of pandemics that threaten worldwide public health due to human activity is often considered to be inevitable [1,2]. The COVID-19 (2019–current) pandemic caused by the emergence in Hubei, China, of what has now been identified as Severe Acute Respiratory Syndrome Coronavirus 2/Novel Coronavirus 2019 (SARS-CoV-2/2019-nCoV) by The Coronaviridae Study Group [3], has brought a number of questions regarding its transmission, containment and treatment to the urgent attention of researchers and clinicians. The urgency of such questions has spurred a number of atypical approaches and collaborations between experts of different fields and as such, this study was carried out as part of a “CoronaHack” hackathon event in April 2020 where the authors gained access to genomes and related metadata available at the time (December 2019–April 2020).

Viruses of the Coronaviridae family have long been studied and while there have been great advances in our understanding, each new emergence has brought about its own questions. Coronavirus consists of four genera: *Alphacoronavirus* (Alpha-CoV), *Betacoronavirus* (Beta-CoV), *Gammacoronavirus* and *DeltaCoronavirus*. Coronaviruses are a group of single-stranded, enveloped and extremely diverse RNA viruses which are known to have come into contact with humans numerous times over the past few decades alone [4]. At around 30 kb, they exhibit at least six Open Reading Frames (ORFs), with ORF1a/b comprising of approximately 2/3 of the genome which encodes up to 16 non-structural replicase proteins through ribosomal frame-shifting, and four structural proteins: membrane (M), nucleocapsid (N), envelope (E) and spike (S) glycoprotein [5]. Coronaviruses have developed a number of different strategies to infiltrate their host cells. In human-associated CoVs, it has been shown that different parts of the human Angiotensin Converting Enzyme 2 (ACE2) can be bound to by their respective S proteins. Pathogens such as SARS-CoV-1 (Severe Acute Respiratory Syndrome Coronavirus) and MERS-CoV (Middle East Respiratory Syndrome Coronavirus) have shown Coronaviruses to be capable of presumed efficient adaptation to their human host and exhibit high levels of pathogenicity [6,7]. Interestingly, SARS-CoV-1 and MERS, which along with SARS-CoV-2 are both Beta-CoVs, exhibit only 79.5% and 50% sequence similarity, respectively, at the whole genome level to SARS-CoV-2, whereas SARS-CoV-2-like coronaviruses found in pangolins (pangolin-CoVs) and bat coronavirus (bat-CoV) RaTG13 (bat-RaTG13) are 91.02% and 96%, respectively [8]. The relationship of SARS-CoV-2 to other SARS-like coronaviruses, the possible role of bats and pangolins as reservoir species and the role of recombination in its emergence are of great interest [9]. Speculations around other intermediary hosts are also at play, which might have affected the ability for zoonotic transmission for SARS-CoV-2 to its human host [10]. Crucially, this evolutionary relationship between SARS-CoV-2 and its lineage may prove to be an important factor in the eventual management or containment of the virus. Moreover, the mutation events along the evolutionary timeline of SARS-CoV-2 are of importance in the discovery of possible adaption signatures within the viral population. At the time of the hackathon, there were two main suspected SARS-like reservoir host species: bat and pangolin (named bat-CoV and pangolin-CoV).

With this in mind, our study aimed to systematically compare a broad selection of contemporary available SARS-CoV-2, bat-CoV and pangolin-CoV at genome, gene, codon usage and variant levels, without preference for strains or sub-genera. This was comprised of 46 SARS-CoV-2 genomes isolated early in the pandemic from Wuhan, China (Late 2019–Early 2020); 117 SARS-CoV-2 genomes isolated in Germany, representing the later stage of global transmission; 215 bat-CoV genomes of Alpha-CoVs and Beta-CoVs; and seven pangolin-CoV genomes, of which five were annotated as Beta-CoVs. During the hackathon, it was recognised that potential biases can arise from directly comparing SARS-CoV-2 to a wide repertoire of coronaviruses of varying stages of genome annotation. Therefore, we performed a new comparative annotation of all genomes used in this study. To further validate mutational adaptations which may have facilitated the zoonotic transmission of SARS-CoV-2, a codon usage analysis was carried out between the SARS-CoV-2 reference genes and the genes identified using the aforementioned approaches. In addition, we profiled codon usage bias across our data set, as in the process of host adaptation, viruses can evolve to express different preferential codon usages [11,12,13].

Through examining the inherent sequence diversity between a comprehensive collection of SARS-CoV-2, bat-CoV and pangolin-CoV, we aimed to highlight naturally occurring high impact variations that can potentially introduce a change in the resulting protein, such as the insertion or deletion of an amino acid or early termination of the sequence. Understanding the stability and variability of these positions may potentially aid future design of vaccines or treatments. For instance, an amino acid position where insertion or deletion is commonly found in a coronavirus affecting other species may indicate that its alteration does not have a dramatic impact on the overall protein folding, or that the position is important for transmission to a new host.

Our work is differentiated by the way of a systematic approach was used to process a non-selective group of these viral genomes from public repositories, prior to applying a wide range of contemporary methodologies and genomic knowledge that highlight the variations that exist between different host species. Understanding the current limitations of annotation pipelines and available curated SARS-CoV-2 genomes was the main driver of this approach. Providing a comprehensive gene and variant annotation for viral genomes collected from multiple hosts will bridge this knowledge gap in the literature.

## 2. Results

### 2.1. Data Collection and Phylogenetic Analysis

We were able to collate 215 bat-CoV genomes of varying families (Alphacoronaviruses and Betacoronaviruses) with only one exhibiting a small proportion or genomic uncertainty (presence of 0.45% “N” nucleotide). However, only seven pangolin-CoV genomes, of which five were annotated as Betacoronaviurs, were available at the start of this study. Three pangolin-CoV genomes also contained levels of the ambiguous “N” nucleotide, two of them at high levels (6.88 and 8.19%). A population of post-outbreak SARS-CoV-2 genomes from Charite [14], Germany, were also collated for further analysis. For the phylogenetic analysis, we examined the complete set of 269 genomes (seven pangolin-CoV; 47 SARS-CoV-2, including the reference genome; and 215 bat-CoV). The phylogenetic tree produced at the whole genome level showed a clear separation between the SARS-CoV-2 and the bat-CoV genomes, with the exception of bat-RaTG13 which has been placed adjacent to the SARS-CoV-2 clade (Figure 1). The similarity of bat-RaTG13 to SARS-CoV-2 has previously been reported [15]. While more distantly related to SARS-CoV-2 than bat-RaTG13, MG772933 and MG772934 (bat-SL-CoVZC45 and bat-SL-CoVZXC21 isolates) are more closely related to SARS-CoV-2 than the remaining bat-CoV (Figure 1). Six of the seven pangolin-CoV genomes are grouped together and closest to the SARS-CoV-2 clade, other than bat-RaTG13. One pangolin-CoV, MT084071.1 (MP789 isolate; referred to as pangolin-MP789), is placed in a branch closer to SARS-CoV-2 than the remaining pangolin-CoV in the tree (Figure 1). The tree produced was used as an analytical anchor for which we could use to refer to in the results from variant analysis. High impact variants were annotated on the tree to show their distribution across the different clades along the topology of the tree.

### 2.2. Gene Identification

For each viral genome, a complementary approach using both PROKKA [16] and BLAST [17] was employed for identifying genes highly similar to those in the SARS-CoV-2 reference genome released by Ensembl v100 (SARS-CoV-2 ref). The breakdown of this result is shown in Table 1, and Table A1 presented a detailed account of the genes annotated in each genome and their corresponding annotation tools (PROKKA or BLAST).

PROKKA, which is an alignment-free method, was unable to capture some genes in some of the genomes; BLAST-alignment was used to address this. This has enabled the characterisation of E and ORF10 in many genomes. Genes utilising ribosomal frame-shifting, such as the aforementioned ORF1ab, are inherently difficult to identify correctly without extensive analysis involving techniques and evidence such as RNA expression analysis. For the majority of genomes studied, PROKKA was able to identify two large ORFs spanning almost the entire length of the ORF1ab locus and detect a central coronavirus frame-shifting stimulation element (named Corona_FSE and separating the two ORFs) which is a conserved stem-loop of RNA found in coronaviruses that can promote ribosomal frame-shifting [18]. The gene sequences generated by PROKKA and BLAST (E and ORF10) were used for downstream analysis, including gene–gene network graph, codon usage bias analysis and a gene presence summary table. The gene presence summary table notates whether SARS-CoV ref genes were found (≥80% percentage identity and ≥50% sequence coverage) in each genome; this table is available in the GitHub project https://github.com/coronahack2020/final_paper/tree/master/host-data.

### 2.3. Gene Relationship Network Graph

A gene–gene similarity network analysis was used to compare genes across SARS-CoV-2, bat-CoV and pangolin-CoV. The advantage of using a 3D network approach to visualise this information was that it simplifies complex information as patterns. Genes sharing high similarity form independent clusters. In cases where there is a high degree of dissimilarity in a gene for different host species, a pattern of 2 or more distinct clusters would take place, with each cluster comprised of genes derived from samples of the same host species. In genes where there is a medium level of dissimilarity across host species, two or more cluster would appear fused and potentially break apart into distinct clusters if the edge threshold were increased. Both of these patterns are observed within this dataset. Distinct separation by host species are seen in ORF1a, ORF3a, ORF6, ORF7a, ORF8 and S (Figure 2). The strongest host–species separation observed was between SARS-CoV-2 and bat-CoV; pangolin-CoV always grouped closer to SARS-CoV-2 than to bat-CoV, with the exception of bat-SL-CoVZC45, bat-SL-CoVZXC21 and bat-RaTG13. In the cases of ORF3a, ORF8 and S, complete separation was observed between bat-CoV and human SARS-CoV-2 (Figure 2B,C). Bat-RaTG13 was more similar to SARS-CoV-2 and pangolin-CoV than the remainder of the bat-CoV for S (Figure 2C). For ORF3a, bat-SL-CoVZC45, bat-SL-CoVZXC21 and bat-RaTG13 clustered together with SARS-CoV-2 and pangolin-CoV rather than with the remainder of the bat genomes (Figure 2). These same three genomes are the only bat-CoV with ORF8 that co-cluster with SARS-CoV-2 ORF8 under the percentage identity threshold (≥80%) set for building the network graph. Other bat-CoV ORF8 were so distinct from SARS-CoV-2 ORF8 that they do not form edges with SARS-CoV-2 ORF8. Interestingly, within the cluster of ORF8 sequences, the ORF8 for pangolin-MP789 shares an average of 92.14% identity to SARS-CoV-2 ORF8, while the ORF8 for remaining pangolin-CoV do not share a strong similarity to the SARS-CoV-2 ref ORF8 (no BLAST result). An average percentage of identity between SARS-CoV-2 ORF8 and bat-CoV ORF8 are 97.05% (bat-RaTG13) and 88.58% (bat-SL-CoVZC45 and bat-SL-CoVZXC21).

To investigate whether if potential gene transfer or recombination that may have come from more distantly related bat-CoV, we sought for unusual co-clustering between genes characterised from bat-CoV and SARS-CoV-2. We did not observe such pattern; bat-RaTG13 co-cluster with SARS-CoV-2 for many genes and is also the most similar bat-CoV to SARS-CoV-2 at a genome level. Two additional genes identified by PROKKA, Corona FSE, a non-coding frame-shift stimulation element within ORF1ab and s2m, a stem-loop II-like motif [19] have both been shown to be highly conserved and important for SARS-2-like coronaviruses. s2m has been identified as a mobile genetic element which has been described in a number of single-stranded RNA virus and insect families and has also been shown to be important for viral function [20,21].

In summary, the use of gene–gene network analysis enables us to determine groups of closely related genes, which not only highlights genes showing strong host–species separation, but also characterise clusters of related genes that may be absent or highly different from the reference genome of interest, such as ORF8. Six genes—ORF1ab, ORF3, ORF6, ORF7a, ORF8 and S—showed a strong host–species separation in the network graph. In particular, with the exception of S, where bat-SL-CoVZC45, bat-SL-CoVZXC21 clustered closer to bat-CoVs, the bat genomes, bat-SL-CoVZC45, bat-SL-CoVZXC21 and bat-RaTG13, clustered together with SARS-CoV-2 than the remainder of the bat-CoV for these 5 genes.

### 2.4. Codon Usage Bias

We examined Relative Synonymous Codon Usage (RSCU) across SARS-CoV, bat-CoV and pangolin-CoV for each SARS-CoV-2 reference gene. Principle component analysis (PCA) using RSCU showed a strong host–species separation; the first principle component (PC1) accounts for 55.62–85.38% of variation (Figure 3), predominately separating SARS-CoV-2 from bat-CoV. Bat-RaTG13, bat-SL-CoVZC45 and bat-SL-CoVZXC21 and pangolin-CoV are usually placed between SARS-CoV-2 and other bat-CoV. With the exception of ORF7b, Pangolin-MP789 is placed closer to SARS-CoV-2 than all other pangolin-CoV (Figure 3) with regards to the variation described by PC1 and PC2.

K-means clustering was used to group the genomes into three clusters for each gene using the first 10 PCs, which have grouped pangolin-MP789 with SARS-CoV-2 for ORF1a, ORF8, ORF7a, E, ORF6 and N (one of two assemblies) Figure A3. For M and ORF3a, pangolin-MP789 clustered with bat-SL-CoVZC45 and bat-SL-CoVZXC21 Figure A3.

A summary of the synonymous codon ratios (the number of codons divided by the total number of codons coding for the same amino acid), sorted by amino acids, are shown in Figure A4.

### 2.5. Variant Analysis

Haplotype-aware variant calling and variant effect prediction of all genomes in the study have been summarised in Figure 4 and Appendix A. There is a total of 1127 variants that are missense, inframe deletion, inframe insertion, stop gained and stop lost, as can be seen in Figure A5. We have removed missense from further analysis and came to a total of 24 high impact variations in eight genes were when comparing bat-CoV and pangolin-CoV genomes against the SARS-CoV-2 ref. We have annotated the majority (with the exception of the NC045512_27675A>ACAG) of these variation in Figure 1, and found that some of these variations, such as variants identified in E, ORF7a and ORF3a, appear to exhibit some degree of clade specificity. The only stop gain variant (i.e., NC045512_29635) was present in ORF10 gene of 57 bat-CoV genomes (29,635 bp position C > A) which was only representing a synonymous variant in the same position of six pangolin-CoV genomes. This variant affected 26Y > 26* (Tyrosine to STOP codon TAC > TAA) in bat ORF10. Assuming the direction of host selection from bat and pangolin to human, this variant could explain the presence of a longer ORF10 isoform in the two latter hosts in comparison to bat-CoV. From the list of variants in Figure 4, four in-frame insertions were identified as follows:ORF1ab gene at position 9757 (NC045512_9757 T > TAGA 3164R > 3164RR) of all pangolin-Cov genomes which represents an extra Arginine.E gene at position 26448 (NC045512_26448 T > TGAA 68S > 68SE) in 33 bat-Cov genomes which caused an addition of Glutamine.ORF7a gene at position 27672 (NC045512_27672 T > TCAC 93V > 93VH) in 24 bat-Cov genomes by addition of an Histamine.N gene at position 28293 (NC405512z_28293 A > AACC 7Q > 7QP) in 13 bat-Cov genomes by addition of a Proline.

Two in-frame deletions were also identified in ORF3a and M genes. A single Glutamine deletion in ORF3a at position 26,111 was present in 14 bat-Cov genomes (NC045512_26111 CTGA > C 240PE > 240P) and a Serine deletion in M gene at position 26,530 (NC045512_26530 ATTC > A 3DS > 3D) was present in 57 bat-Cov genomes. The same position showed a missense mutation of 3D > 3A (in two bat-Cov [bat-SL-CoVZC45 and bat-SL-CoVZXC21] and one pangolin-Cov) and 3D > 3G in six pangolin-Cov genomes.

## 3. Discussion

During the 5-day hackathon, we endeavoured to utilise the genomic data aggregated by the scientific community and undertook a multifaceted and comprehensive exploration of the genomic sequences (or “similarities and differences”) of coronaviruses infecting bat and pangolin hosts, available at the time. We have compared SARS-Cov-2 to all bat-CoV and pangolin-CoV genomes from the listed data repositories (NCBI, VIPR and Databiology) without selecting for strains to represent any specific genera, species or substrain. Our comparisons spanned across several levels: whole-genome, genes, codons and individual variants.

The origin of SARS-CoV-2 is still unknown and a number of coronaviruses from different hosts have been proposed as the potential common ancestors [22,23]. However, bats are often linked to SARS-like viruses capable of zoonotic host transfer due to their unique niche as viral reservoirs. This is often characterised by their physiology relatively unaffected under varying viral loads and their natural proximity to human habitation [24,25]. Furthermore, recombination has been suggested as an avenue for host transfer for a number of RNA viruses such as SARS-CoV-1 and MERS [26,27].

The phylogenetic tree inferred from genomes studied in this manuscript presents a picture of vast bat-CoV diversity and its topology is similar to those of previous studies carried out on pangolin and bat coronaviruses when compared to the SARS-CoV-2 genome [28]. Previous phylogenetic profiling has noted that bat-RaTG13 bares the closest resemblance to SARS-CoV-2 across 55 SARS-like coronavirus genomes [29]. Of the the 222 SARS-like coronavirus genomes we have constructed the phylogenetic tree with, bat-RaTG13 remains the closest to SARS-CoV-2, followed by pangolin-MP789, the remaining six pangolin-CoV, and then bat-SL-CoVZC45 and bat-SL-CoVZXC21. The relationships between pangolin-MP789 and the three aforementioned bat-CoVs have been described [30], but it has not yet been highlighted that pangolin-MP789 is closer to SARS-CoV-2 than the other known pangolin-CoV (Figure 1). This relationship has previously been reported and a recombination event between pangolin-CoVs and bat-RaTG13 has been theorised [31].

As well as at genome level, the similarity of bat-RaTG13 and pangolin-MP789 to SARS-CoV-2 is also evident at gene level, in particular, across ORF8 sequences. Only a few closely related SARS-CoV-2 ORF8 orthologues have been identified within bat-betacoronavirus lineages [32,33]. We have shown the pangoling-MP789 and bat-RaTG13 ORF8 gene has ≥90% sequence identity to the SARS-CoV-2 ref ORF8. The exact function of ORF8 remains to be elucidated, although studies on ORF8 from SARS-CoV-2 and ORF8ab and ORF8b from SARS-CoV-1 have suggested a role in immune modulation through the interferon signalling pathway [34,35] and inducing strong antigen response [36]. Although the origin or function of the SARS-related coronavirus ORF8 remains unresolved, a 29-nucleotide deletion in ORF8 is often found in SARS-CoV-1, when compared to civet-CoV, suggesting that ORF8 may be important for interspecies transmission [37].

Other genes that show strong host-species separation in the gene–gene network analysis include ORF1a, ORF3a, ORF6 and S. It has been previously shown that pangolin-CoV and SARS-CoV-2 S proteins were highly similar to each other (97.5%) [38]. Furthermore, it has been shown that the overall structure of S protein in bat-RaTG13 is highly similar to those in SARS-CoV-2 [39]. This is significant as the S protein plays an important role in the initial penetration and infection of host cells and are often host-specific [40]. Viruses, through co-evolution with the host have high degrees of flexibility in their receptor usage and capacity to reach binding efficiencies via mutations [41,42] Several human coronaviruses, including SARS-CoV-2, SARS-CoV-1 and human coronavirus NL63 (hCoV-NL63), penetrate the host cell by binding to the host ACE2 through the receptor binding domain (RBD) of S protein [43,44]. It would appear that despite the S protein being more similar between pangolin-CoVs and SARS-CoV-2, the S protein in bat-RaTG13 is still more similar to that of SARS-CoV-2 than other bat-CoVs in our study (Figure 2C). This raises the possibility that the most recent common ancestor of SARS-CoV-2 (be of pangolin-CoV or bat-CoV origins) is yet to be sequenced.

Codon usage bias across the species–host range may show signs of preferential codon mutation which have occurred during the complex process of host interaction and transfer [11,12]. The knowledge of nucleotide profiles and subsequent codons during the human–virus co-evolution could be invaluable to the design of vaccines and their continuous development over the years to come [45]. On the whole, the codon usage profiles are highly different between SARS-CoV-2 and the majority of bat-CoV, with bat-RaTG13, bat-SL-CoVZC45, bat-SL-CoVZXC21 and panolin-CoV positioned between the two groups. Similar to the analysis by Gu et al. (2020), we found the codon usage profiles in bat-RaTG13 to be most similar to SARS-CoV-2 on the whole [46]. However, we have included six additional pangolin-CoV isolates in our studies and found pangolin-MP789 exhibited consistently more similar codon usage profiles to SARS-CoV-2 than the remaining pangolin-CoV at the gene level, which is also reflected in the genome-level phylogenetic tree. These observations highlighted the variation within pangolin-CoV and the closer resemblance between pangolin-MP789 and SARS-CoV-2; pangolin-MP789 is an isolate collected in 2019, whereas all other pangolin isolates were collected prior to 2019. Our codon usage analysis has focused on the overall comparison of RSCU for each gene across bat-CoV; other studies have compared gene sequence characteristics such as GC content and CpG dinucleotide [47,48,49].

Next, we focused on variants that could potentially have a more profound impact on the amino acid substitution or early stop codon gains (i.e., truncation). Population-level viral mutation is a complex process, involving a number of pressures, and while RNA viruses often exhibit some of the highest mutation rates of all viruses, conserved variants can exhibit important functional changes such as the ability to evade immunity more efficiently [50]. Furthermore, unlike the vast majority of RNA viruses, coronaviruses encode a complex RNA-dependent RNA polymerase that has a 3’ exonuclease domain [51], effectively proofreading mutational events and therefore are less error-prone. Therefore, the mutations observed across populations have undergone an error-correction process which means they are more likely to be functionally beneficial to the virus.

We have observed several of such variants (allele frequencies > 0.1) that are at consistent loci across different bat-CoV clades as shown in Figure 1. Some of these variants are seen in the majority of the bat-CoV samples (which align to the SARS-CoV-2 ref), including a stop-gain for ORF10 and an in-frame deletion for M, whilst others, such as the variants seen in ORF7a and E appear to be clade specific (Figure A1). Several of these variants affect the same amino acid positions, including E (in-frame insertion of *Asp* (Aspartic acid), *Glu* (Glutamic acid) or *Gln* (Glutamine) at at positions 68), N (inframe insertion of *Pro* (Proline) or *Ser* (Serine) at position 7) and ORF7a (in-frame insertion of *His* (Histidine), *Gln* or *Tyr* (Tyrosine) at position 93) (Figure A1). Notably, the stop-gain was identified at amino acid position 26 in ORF10 for 57 of the 59 bat-CoV genomes with ORF10 that had ≥80% similarity to the SARS-CoV-2 ref. The absence of this stop codon in the pangolin (which exhibited synonymous mutations at the same locus) and SARS-CoV-2 viruses could result in a longer isoform of the ORF10 or fundamental changes in its function and expression levels. In a previous study of SARS-CoV-2 and pangolin-CoV genomes, position 26 was also identified as a region of population level variation from *Tyr* and *His* which significantly modifies the secondary structure of the coil region of the protein [52].

There has been little research on ORF10 function, and its expression has been the subject of debate. Whilst Kim et al. (2020) found little evidence of ORF10 expression (0.000009% of viral junction-spanning reads) in cell culture (Vero cells) [53], Liu et al. (2020) found it to be abundantly expressed in severe COVID-19 patient cases but barely detectable in moderate cases [54]. Besides the single ORF10 variant that is observed in the majority of the bat-CoV, we have observed three different amino acid insertions (four different nucleotide changes) at position 68 of E gene in four different clades of bat-CoVs.

The small envelope E protein is the smallest of coronaviruses’ major structural proteins, but also one of the least described [55]. E gene has been shown to be highly expressed inside infected cells and the viruses which are formed without E exhibit reduced levels of viral maturation and tropism. Expression of the E product was essential for virus release and spread, thus demonstrating the importance of E in virus infection and therefore vaccine development [56]. The 68th amino acid position we highlight in this study is in the c-terminal domain, which coincides with the previously reported motif in SARS-CoV-1 (also at 68th amino acid position) that binds to the host cell PALS1 protein to facilitate infection [57]. Less than 0.5% of 3617 SARS-CoV-2 genomes have been found to have non-synonymous mutation in E, and of these, 20% are at the 68th amino acid position [58]. These changes in amino acid may alter the hydrophobicity at the locus, thus possibly influencing the protein functions and interactions [58]. Two of the E variants we highlighted use different codons for the same amino acid (GAG or GAA for *Glu*), which potentially suggests interplay between the selection pressures of codon optimisation and amino acid insertion into the protein product.

We have characterised a number of in-frame insertions at the amino acid position 93 in ORF7a across 55 bat-CoV genomes, and at position 94 reported in two. As with position 68 in E, position 93 in ORF7a has multiple codon insertions coding for the same amino acid but in two groups. In these two groups of bat-CoVs, an additional *His* is encoded for by two different codons and secondly, so is *Tyr* in another group. Intriguingly, ORF7a in SARS-CoV-1 has been shown to regulate the bone marrow stromal antigen 2 which inhibits the release of virions of human-infecting viruses [59].

N is another gene for which we have shown multiple in-frame insertion variants for the same amino acid position. The N protein is highly expressed during an infection, and it plays a key role in promoting viral RNA synthesis and incorporating genomic RNA into progeny viral particles [60]. In gene N, we observed two in-frame insertions at amino acid position 7 for *Ser* or *Pro* from two groups of bat-CoVs (13 and 11 respectively), as well as two in-frame deletions at positions 238 and 385. For M in 57 bat-CoV and pangolin-CoV, there is an in-frame deletion at position 3, which removed the amino acid *Ser*. At this amino acid position, a missense mutation of (Asp) to Glycine (Gly) is seen in 2 bat-CoV (bat-SL-CoVZC45 and bat-SL-CoVZXC21) and pangolin-MP789, and (Asp) to *Arg* in the remaining 6 pangolin-Cov genomes. Bat-SL-CoVZC45, bat-SL-CoVZXC21 and pangolin-MP789 have been shown to be more similar to SARS-CoV-2 than other coronavirus of the same host on other comparative metrics. M plays an important role in its interactions with both E and S to incorporate virions into the host cells.

The amino acid positions we have highlighted through our variant analysis may constitute important differences in the function or folding potential of the protein product. We have summarised the polymorphism along with respective allele frequencies and amino acid consequences in Figure 1. Weber et al. (2020) have interrogated 572 SARS-CoV-2 genomes isolated worldwide and characterised 10 distinct mutation hotspots that have been found in up to 80% of the viral genomes they examined [61]. While our reported variant positions are not 100 % in concordant with these hotspots, some of them display changes on or adjacent to our reported positions.

Through employing a number of genomic analysis methodologies, this study has aimed to bring understanding of the diversity across SARS-CoV-2 and SARS-CoV-2-like coronaviruses by comparing a wide selection of available genomes from the (early stages) starting point of the pandemic. We have highlighted a high degree of host-species separation in sequence homology for ORF3a, ORF6, ORF7a, ORF8 and S, as well as codon usage. Along with bat-RaTG13, we have highlighted the pangolin-MP789 isolate to bare stronger resemblance to SARS-CoV-2 than other pangolin-CoV in both whole-genome phylogenetic tree and gene-level codon usage profiling. Furthermore, a number of amino acid positions that demonstrate high impact variants (inframe insertion/deletion or stop gain) have also been identified in various bat-CoV and pangolin-CoV; these are potentially functionally important positions that warrant further research. The as-yet unknown evolutionary road map undertaken by the ancestor of SARS-CoV-2 to cross over to its now human host is to be investigated for understanding its origin.

## 4. Methods

### 4.1. Genomes

Historically, genomes held in public databases have been fragmentary, resulting in multiple collections with overlapping examples with alternative naming schemes and annotations. Fortunately, a large collection of virus genomes of the Coronaviridae family (Coronavirus) deposited in databases such as the Virus Pathogen Resource (ViPR) [62] have been provided with both genomic sequence and metadata which has been examined for redundancy and comparative annotation. Coronavirus genomes isolated from humans, bats and pangolins used in this study were collected from multiple repositories and grouped by their host and source. The databases and groups are listed in Table 2.

### 4.2. Genome Annotation

RNA viruses such as SARS and other coronaviruses have been characterised as having the ability to utilise ribosomal programmed frame-shifting for a number of important genes [65]. Identification of such genes is complex and often requires high quality RNA expression evidence. Due to this and the complexity of genome annotation, especially in novel viral genomes such as SARS-CoV-2, two approaches were taken to identify the set of genes for each of the genomes in this study. In this regard, for defining genes, we first employed PROKKA (Rapid Prokaryotic Genome Annotation) to curate the genes for each of the coronavirus genomes. PROKKA utilises Prodigal [66] to initially find ORFs, which ensures that the DNA sequences of the genes found are in-frame and contain the correct amino acid coding potential. Prodigal is an unsupervised ab initio prediction method and therefore does not rely on previous knowledge to predict ORFs, which, unlike sequence homology based tools such as BLAST, does not require previously annotated sequence data to identify potential genes within novel genomes. However, to overcome the limitations and intricacies of contemporary *ab initio* genome annotation techniques, BLAST was used to identify additional genes with strong homology to those present in the SARS-CoV-2 reference genome released by Ensembl v100 (SARS-CoV-2 ref) *ASM985889v3* [63] (https://covid-19.ensembl.org). The additional BLAST annotation was performed with a BLAST percentage identity threshold of ≥80% are labelled separately where annotation methodologies may have an impact. This combined approach was used to avoid solely relying on either method, especially BLAST’s agnostic approach to coding frame detection.

### 4.3. Phylogenetic Trees

A Phylogenetic tree was produced from the genomes of the SARS-CoV-2 Wuhan isolates, Ensembl Wuhan reference and the bat and pangolin coronaviruses to examine their evolutionary relationships at the genomic level. Clustal Omega 1.2.4 [67] was used to perform a multiple sequence alignment for each of the genomes with default parameters. The phylogenetic tree was inferred from the multiple sequence alignment with RAxML [68] using default parameters apart from the GTRGAMMA option and bootstrapping set to 20. The plotted using packages in R. Midpoint-root and ladderized were carried out using phytools [69] and ape [70], and ggtree [71] was used for the visualisation. The subgenus information for *Betacoronavirus* were curated and clades labelled based on consensus of the majority (i.e., if >85% of the samples in the clade are labelled and have the same subgenus annotation). For labelling the bat-CoVs host genera and species information, a list of host genera and species was curated. Host species with >10 bat-CoV genomes were labelled, followed by host genera with more >10 bat-CoV genomes. The remaining bats were grouped into a single group “other”.

### 4.4. Gene Relationship Network Graph

Genes identified by PROKKA from each host set were collated and together with the additional sequences from the BLAST alignment to the SARS-CoV-2 ref genome as aforementioned, an all-against-all comparison was made with BLAST. This was done with all gene sequences as both the reference and the query as input. A network graph was generated using Graphia Enterprise [72] by treating each gene as a node and generating edges between nodes with significant BLAST alignments. A significant BLAST alignment was defined to have a bit score ≥ 60, a query coverage ≥ 80% and a percentage identity ≥ 80%. Components with less than 5 nodes were removed from the graph. The same procedure was carried out using amino acid sequences as input (Figure A2). Where the amino acid sequences were not generated by PROKKA, the matched sequences extracted from BLAST were translated into amino acid sequences, provided that the sequences contained the start and stop codons.

### 4.5. Codon Usage

Codon usage metrics for every gene in the SARS-CoV-2 reference gene catalogue were calculated in all available genome sets. Gene sequence output of the PROKKA and BLAST searches (where correct frame was present) were collated and BLAST searched against the SARS-CoV-2 ref genes; genes that have a BLAST result were included and annotated with the SARS-CoV-2 gene. For each set of genes annotated with an SARS-CoV-2 gene, those substantially shorter than the average (<mean length—2 standard deviation) were removed from codon usage analysis.For ORF6 and ORF8, the BLAST filter criteria yielded few bat-CoV (11 and 3) or pangolin-CoV (1 and 6) genes. Therefore, in addition to the BLAST selected genes, bat-CoV and pangolin-CoV genes labelled as ORF8 and ORF6 in the network analysis (Figure 2) were incorporated in the codon usage analysis. For pangolin-MP789, the PROKKA output from an additional assembly (MT121216.1) was included in the codon usage analysis. Custom Python scripts (available on Github (https://github.com/coronahack2020/final_paper.git) were used to summarise the frequencies of each of the codons for each gene. Non-standard codons, start and stop codons were discarded, along with the codon TGG as it is the only codon coding for tryptophan. PCA was performed on the RSCU values, and kmean clustering was used on the first 10 PCs to group the genomes into 3 clusters.

RSCU was calculated as the ratio of the observed frequency of codon to the expected frequency under the assumption of equal usage between synonymous codons for the same amino acids [73].

### 4.6. Variant Analysis

For this analysis, we aim to highlight naturally occurring and population-wide viable variants, defined as being different to the SARS-CoV-2 ref and have an impact on coding potential. Variant calling was carried out for all available genome sets against the reference SARS-CoV-2 genome released by Ensembl v100 *ASM985889v3*. The allelic counts and variant effect prediction was carried out in order to identify variants with high impact changes (inframe deletion, inframe insertion, frameshift, or stop gain) within or between viruses collected from different host species.

Briefly, multiple genome fasta input files were mapped against the SARS-CoV-2 ref assembly using minimap2 [74] with the following flags (*minimap2–cs-cx asm20 INPUT REF > OUT.paf*). The generated PAF (pairwise alignment format) files were subsequently used for variant calling through the paftools.js module in minimap2 (*sort-k6,6 -k8, 8n OUT.paf|paftools.js call-l 200-L 200-q 30 -f REF.fa*). Haplotype aware variant consequences were generated using VEP (Variant Effect Predictor) [75,76]) and BCFtools/csq [77]. The complete set of scripts for this pipeline can be found in https://github.com/coronahack2020/final_paper.git.

## Figures and Tables

**Figure 1 viruses-13-00049-f001:**
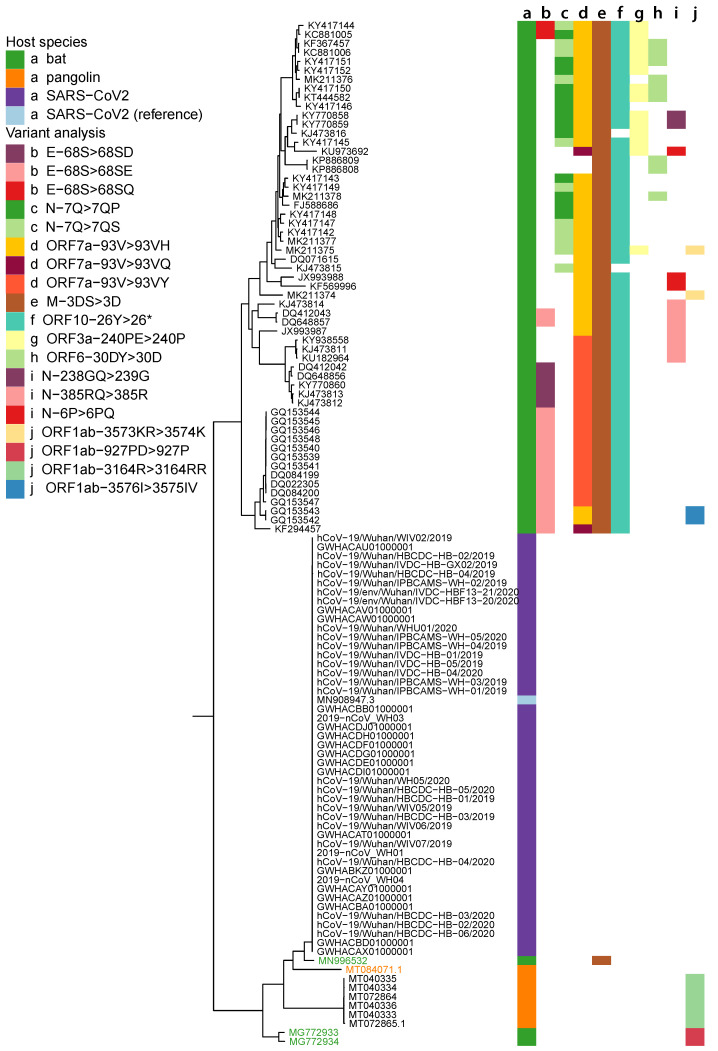
Phylogenetic tree showing relationship between bat-CoV, pangolin-CoV and SARS-CoV-2. This is the Sarbecovirus clade from Figure A1, the phylogenetic tree made with all bat-CoV, all pangolin-CoV and SARS-CoV-2 (Wuhan dataset and SARS-CoV-2 reference) used in this study. Along with the (a) host organisms, results from the variant analysis are annotated, showing (b–d) positions with multiple amino acid changes, (e–h) positions with a single amino acid change (in >10 genomes) and (i,j) other variants. The genes and amino acid changes involved in each of the annotated inframe insertion, inframe deletion or stop gain (*) are indicated in the figure legend. The names of four genomes are highlighted, including 3 bat-CoV—MN996532 (bat-RaTG13), MG772933 (bat-SL-CoVZC45) and MG772934 (bat-SL-CoVZXC21)—and 1 pangolin-CoV, MT084071.1 (pangolin-MP789), as they are more closely related to SARS-CoV-2 than the other bat-CoV or pangolin-CoV in the tree.

**Figure 2 viruses-13-00049-f002:**
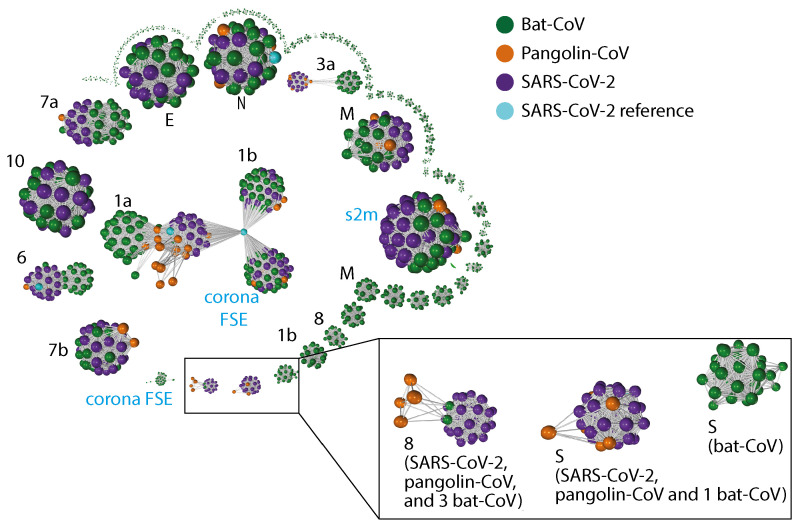
Gene–gene similarity network analysis. Each node represents a gene defined by PROKKA or a DNA segment similar to genes from the SARS-CoV-2 reference genome. The nodes were compared against each other using BLAST, and nodes with high similarity (bit score ≥ 60 and a query coverage ≥ 80%) were connected with an edge. The network graph is labelled with host species. The black font in the graph indicates the corresponding SARS-CoV-2 gene names (“Open Reading Frame (ORF)” omitted) for the larger clusters, whereas blue font indicate additional non-coding sequences defined by PROKKA. Instead of the full length ORF1ab ( 21kb in length), ORF1a and ORF1b were defined by PROKKA as two separate genes. Notably, ORF1a, ORF3a, ORF6, and ORF8 and S show strong separations between nodes from different species. ORF8 from 3 bat-CoV co-clusters with ORF8 from SARS-CoV-2 (bat-RaTG13, bat-SL-CoVZC45 and bat-SL-CoVZXC21, respectively). The remaining bat-CoV ORF8 do not co-cluster with SARS-CoV-2 ORF8 even without the edge filtering threshold. For S, the bat-RaTG13 co-cluster with COVID-19 and pangolin. A cluster of bat-CoVs break off for ORF1b and M, suggesting a large amount of variation amongst bat-CoV for these genes.

**Figure 3 viruses-13-00049-f003:**
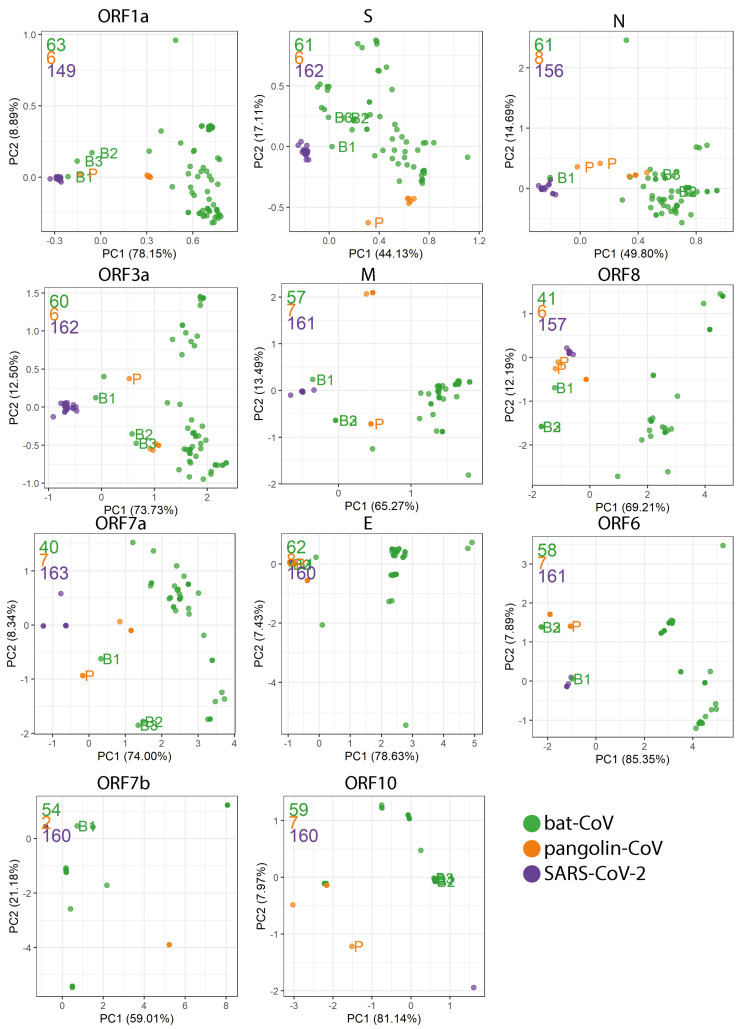
Relative synonymous codon usage (RSCU) was calculated as the ratio of the observed frequency of codon to the expected frequency under the assumption of equal usage between synonymous codons for the same amino acids. For each gene, Principal Component Analysis (PCA) was carried out on the RSCU values. The first two Principal Components (PCs) are plotted. The total number of genomes used in each plot are indicated in the top left corner in the corresponding colour. In order, they are bat-CoV (green), pangolin-CoV (orange) and SARS-CoV-2 (purple). Four isolates are labelled: bat-RaTG13 (B1), bat-SL-CoVZC45 (B2), bat-SL-CoVZXC21 (B3) and pangolin-MP789 (P; MT121216.1 and MT084071.1).

**Figure 4 viruses-13-00049-f004:**
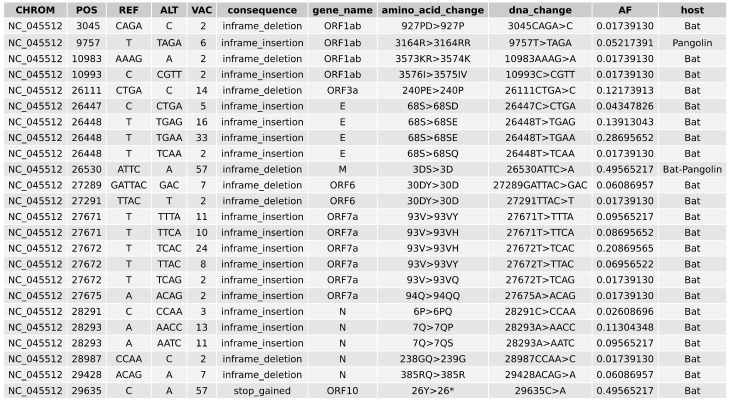
High impact variants identified across bat and pangolin genomes using the variant calling pipeline based on SARS-Cov-2 Ensembl reference genome. The variants with allele frequency > 0.1 and predicted to have HIGH impact using VEPTools are listed. CHROM: Reference contig name; POS: Position; REF: Reference allele in Ensembl Human SARS-Cov2; ALT: Alternative allele(s) found in non-human genomes; VAC: Alternative variant allele counts; AF: Allele frequency.

**Table 1 viruses-13-00049-t001:** This table presents the distribution of the number of predicted genes for each dataset. Bat-CoV exhibit the widest distribution of gene count, and pangolin-CoV has the highest number of gene count, with one genome having 17 predicted genes. These outliers have low sequence or assembly quality. In the case of the pangolin-CoV genome reporting 17 genes, it has low-quality (“NNNN”) nucleotide regions spanning the centre of genes, which causes PROKKA to identify the two ends of one gene. The median gene count only varying in bat-CoVs, likely attributed to the large phylogenetic variation exhibited across the bat-CoVs.

Dataset	Min.	Median	Mean	Max.	Sample Count
Wuhan	7	11	11	13	46
Charite	9	11	11	12	117
Bat	2	9	9	12	215
Pangolin	10	11	12	17	7

**Table 2 viruses-13-00049-t002:** Coronavirus genomes were collected from the various database resources listed by host and source categories. Using taxonomic data made available by the Virus Pathogen Database and Analysis Resource (ViPR) [62], 70 bat-CoVs were identified as *Betacoronavirus* and 84 were *Alphacoronavirus*. Five pangolin-CoVs were identified as *Betacoronavirus*. The remaining bat-CoV and pangolin-CoV genomes did not have a family identification. These were downloaded in May 2020 and consisted of the contemporary available and open datasets at the time. All genomes and their respective IDs are currently available through NCBI (Oct 2020). In cases where two groups contained the same genome (Possibly with a different name), only one representative was taken.

Host–Source	No. Genomes	Database
SARS-CoV-2 Wuhan isolates	20	https://doi.org/10.1101/2020.10.22.328864
SARS-CoV-2 Wuhan isolates	26	GISAID-Charite [14]
SARS-CoV-2 German isolates	117	GISAID-Charite [14]
SARS-CoV-2 Ensembl Wuhan Reference	1	Ensembl [63]
Bat	139	https://doi.org/10.1101/2020.10.22.328864
Bat	76	ViPR [62]
Pangolin	5	https://doi.org/10.1101/2020.10.22.328864
Pangolin	2	NCBI [64]

## Data Availability

The sources for datasets used in this study are detailed in Table 2.

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
