# Peer review of "Computational Analysis of SARS-CoV-2 and SARS-Like Coronavirus Diversity in Human, Bat and Pangolin Populations"

_viruses, 2020, doi:10.3390/v13010049_

Round 1
Reviewer 1 Report
The author analysed SARS-CoV-2/COVID-19 sequences along with bat and pangolin coronavirus sequences. The authors performed de novo gene annotation, mutation and codon usage bias analysis. Though, some of the analysis are interesting, however they lack novelty in the present form and further analysis need to be performed to make it more novel. Some of the other flaws in the manuscript are discussed below as well:
Major:
- Authors found codon usage bias analysis to correspond exactly as the different clades in the phylogenetic tree. Main reason is that Fig3 has combined multiple genes’ codon usage bias. However, I think each gene usually follows its own unique path for the codon usage bias optimization. Clubbing multiple genes’ data as homogenous mixture is surely going to skew the and obscure the data of individual gene’s codon optimization. Hence, I recommend removing this figure and use the appendix figure with individual genes’ codon usage bias analysis instead.
- As discussed, novelty of results is an issue at this point as many publications on Covid19 have addressed similar points. Gu et al. has performed codon usage bias analysis, identifying spike and membrane genes to be under different evolutionary pressures.
Hence, author should perform some analysis differently here, as they have the pangolin data already. Authors need to focus more on identifying the novel pangolin virus strain which clusters more closely with the Covid19.
Usually, a fixed number of clusters is given as input for k-means clustering. So, authors should explain that in the text and ideally they should just use 3 clusters only as input, one corresponding to each species Co-V (bat, pangolin and covid19). After using just 3 clusters, then just let the machine learning decide which strain in which species (bat and pangolin) and in which gene is close upon clustering with covid19 cluster or which ones not. Hence, this analysis will be able to show the novel results into which exact pangolin strain matches best to the codon usage bias pattern of covid19 and I am sure bat-CoV RaTG13 would be the only bat strain clustering with covid19 cluster. From there the theory of recombination between those 2 strains might be more interesting. This exact pangolin strain clustering with covid19 would add novelty to this paper then, but without this analysis, the results might be less novel.
- Are these in-frame insertion and deletions statistically significant for over-representation? Please use some statistical p-values to signify these few amino acid insertions. Can one or few amino acids really change the viral pathogenesis, it should be explained more in the sections where these results are discussed.
Minor comments:
- line 100-102, need more explanation, it comes earlier in a premature manner. Though some explanation is given later. Those 3 lines keeps reader confused whether the "E, ORF8 and ORF10" were found in all species or which specific species. I suggest moving the names of 3 genes down to the end, after when the explanation is given that in which species and how many strains these genes were found. Can it be incorporated to Table1?
- 'RNAseq expression analysis' should be removed. It could be mentioned in the discussion section, but without any details and confidence in this section, inclusion of this paragraph in results section is not warranted.
- Hacking should be removed from the title. Though I understand, its associated with the start of this project (“CoronaHack” competition), but as a reader of this journal I might not be interested in that and the ‘Title’ could be taken mistakenly as part of results. The hacking has no association with results. So, I recommend title as: Computational analysis of SARS-CoV-2 And SARS-Like Coronaviruses Diversity In Human, Bat And Pangolin Populations
- Authors mention: “pangolin-CoV always group closer to SARS-CoV-2 than to bat-CoV” The authors didn’t mention in the Phylogenetic tree analysis that MN996532 or RaTG13, being closest to Covid19 whole genome sequence. Though it may not be novel anymore.
- For S, the bat-CoV RaTG13 co-clustering with COVID-19 is already known and deeply studied for its effect on binding with human receptor. PMID: 32848232.
- Appendix B. Number of Genes is misleading, maybe use ‘number of sequences matching genes’
Author Response
#################Reviewer1
Comments and Suggestions for Authors
The author analysed SARS-CoV-2/COVID-19 sequences along with bat and pangolin coronavirus sequences. The authors performed de novo gene annotation, mutation and codon usage bias analysis. Though, some of the analysis are interesting, however they lack novelty in the present form and further analysis need to be performed to make it more novel. Some of the other flaws in the manuscript are discussed below as well:
Major:
Authors found codon usage bias analysis to correspond exactly as the different clades in the phylogenetic tree. Main reason is that Fig3 has combined multiple genes’ codon usage bias. However, I think each gene usually follows its own unique path for the codon usage bias optimization. Clubbing multiple genes’ data as homogenous mixture is surely going to skew the and obscure the data of individual gene’s codon optimization. Hence, I recommend removing this figure and use the appendix figure with individual genes’ codon usage bias analysis instead.
As discussed, novelty of results is an issue at this point as many publications on Covid19 have addressed similar points. Gu et al. has performed codon usage bias analysis, identifying spike and membrane genes to be under different evolutionary pressures.
## Thank you for the suggestion. We have now replaced the multiple-gene codon usage analysis with individual-gene codon usage analysis, and updated the figures (Figure 3) and text (line 163-173, line 255-270, line 405-422) accordingly. Our study has included a more comprehensive source of pangolin-CoV (7 pangolin-CoV instead of 1 pangolin-CoV in Gu et al.) and bat-CoV than existing analysis. By doing so, we demonstrate the strong separation between SARS-CoV-2 and a wide spread of bat-CoV, as well as the positioning of the potential intermediate genomes (such as pangolin-CoV and RaTG13). We have also highlighted our novel finding that characterises the divergence of pangolin-MP789 from the remaining pangolin-CoV in lines 265-266, 344-346.
Hence, author should perform some analysis differently here, as they have the pangolin data already. Authors need to focus more on identifying the novel pangolin virus strain which clusters more closely with the Covid19.
## Thank you for your suggestion. We have done that now (lines 265-266, 344-346).
Usually, a fixed number of clusters is given as input for k-means clustering. So, authors should explain that in the text and ideally they should just use 3 clusters only as input, one corresponding to each species Co-V (bat, pangolin and covid19). After using just 3 clusters, then just let the machine learning decide which strain in which species (bat and pangolin) and in which gene is close upon clustering with covid19 cluster or which ones not. Hence, this analysis will be able to show the novel results into which exact pangolin strain matches best to the codon usage bias pattern of covid19 and I am sure bat-CoV RaTG13 would be the only bat strain clustering with covid19 cluster. From there the theory of recombination between those 2 strains might be more interesting. This exact pangolin strain clustering with covid19 would add novelty to this paper then, but without this analysis, the results might be less novel.
## Thank you very much for your suggestion. We have modified the analysis to perform k-means clustering on each gene, and updated the text (line 170-173, line 416-419) and Appendix Figure A3 accordingly. Indeed one of the pangolin-CoV genomes (pangolin-MP789) is placed closer to SARS-CoV-2 in the PCA plot for most genes. Furthermore, this is the only pangolin-CoV with high ORF8 similarity to SARS-COV-2, and is also closer to SARS-CoV-2 than the other pangolin-CoVs in the phylogenetic tree (line 96-97, Figure 1). Pangolin-MP789 was isolated in 2019, whilst the other pangolin-CoV were isolated prior. We have now highlighted this genome in Figure 1 and throughout the text (lines 96-97, 265-266, 344-346).
Are these in-frame insertion and deletions statistically significant for over-representation? Please use some statistical p-values to signify these few amino acid insertions. Can one or few amino acids really change the viral pathogenesis, it should be explained more in the sections where these results are discussed.
## We are providing a frequency landscape of insertion-deletion variants in each host specific clade. These variants are not necessarily evidence of change in pathogenesis or host specific pathobiology of the virus. However; by the virtue of their frequency could be stable in the host population. Specifically with the focus on variants with AF above 0.1 (mainly shown in figure 4)
These variant are mere discovery of potential highly variable spots and codon which are found in wild type coronavirus genomes. Some of these position of interest are reported in other studies Weber et al. (2020) (https://doi.org/10.1016/j.virusres.2020.198170).
These variants were processed using Ensembl VEPtools variant effect prediction principles which provide the potential IMPACT category of the variant as described by https://m.ensembl.org/info/genome/variation/prediction/predicted_data.html. Only variants with HIGH category of impact were kept for the analysis presented in this study.
Minor comments:
line 100-102, need more explanation, it comes earlier in a premature manner. Though some explanation is given later. Those 3 lines keeps reader confused whether the "E, ORF8 and ORF10" were found in all species or which specific species. I suggest moving the names of 3 genes down to the end, after when the explanation is given that in which species and how many strains these genes were found. Can it be incorporated to Table1?
## Thanks for the suggestion and the text were amended to remove confusion (lines 107-109).
'RNAseq expression analysis' should be removed. It could be mentioned in the discussion section, but without any details and confidence in this section, inclusion of this paragraph in results section is not warranted.
## We have now removed the RNA-seq analysis from the manuscript entirely.
Hacking should be removed from the title. Though I understand, its associated with the start of this project (“CoronaHack” competition), but as a reader of this journal I might not be interested in that and the ‘Title’ could be taken mistakenly as part of results. The hacking has no association with results. So, I recommend title as: Computational analysis of SARS-CoV-2 And SARS-Like Coronaviruses Diversity In Human, Bat And Pangolin Populations
## Thanks for the suggestion and the title was changed accordingly
Authors mention: “pangolin-CoV always group closer to SARS-CoV-2 than to bat-CoV” The authors didn’t mention in the Phylogenetic tree analysis that MN996532 or RaTG13, being closest to Covid19 whole genome sequence. Though it may not be novel anymore.
## We have now mentioned that RaTG13 is the closest to SARS-CoV-2 and cited papers that have noted that being the case (lines 89-96). We have also highlighted pangolin-MP789 being the second closest to SARS-CoV-2 (closer than the remaining pangolin-CoV) (line 96-97) .
For S, the bat-CoV RaTG13 co-clustering with COVID-19 is already known and deeply studied for its effect on binding with human receptor. PMID: 32848232.
## Thank you for pointing this out. We have now added the reference (lines 243-244).
Appendix B. Number of Genes is misleading, maybe use ‘number of sequences matching genes’
## Thanks for the suggestion and the appendix is modified accordingly (Appendix B).
Reviewer 2 Report
The manuscript entitled. “Hacking the Diversity of SARS-COV-2 and SARS-Like Coronaviruses in Human, Bat and Pangolin Populations,” focuses on the in-depth genetic analysis of SARS-CoV-2 genomes in various animals species. The analyses produced several notable findings including re-confirming the pangolin-CoV genomes cluster similarly with the Wuhan SARS-CoV-2 isolates, separation in host-associated CoVs in various ORFs specifically between pangolin-CoV and human SARS-COV-2, and identification of multiple variants that may have some important function. Overall, the information is interesting, well-written and important to further our understanding of molecular mechanisms associated and origins of SARS-CoV-2. Minor grammatical edits are suggested below:
Line 61: Correct analyse
Paragraph 73-76: Add a sentence or two about why we need to know and understand differing annotations, SNPs etc.
Line 81-82: Are you using additional genomes that have not been utilized in previous studies?
Line 84: Why did you target Germany in particular?
Line 90: May want to add a sentence or two about previous research associated with RaTG13 and why it can be discounted.
Figure 1: The organization of the legend and figure makes it confusing. Please reorganize for clarity. Additionally,may want to put appendix A as in the main text of the manuscript. It is much easier to follow. As it is, it is difficult to differentiate the color changes in such a small section. Finally, did you analyze the human genomes for codon usage or variants? If there were few variants in the pangolin/human genomes, why do you think that is? Did you compare the number of variants between species?
Line 168: Why was 18% used as a cutoff?
Line 173-175: Could the close clustering of these sequences suggest a similar origin? Or error in the assay?
Line 233-234: From your data, can you suggest a more likely origin?
Discussion: Very long, please shorten. Highlight main take-aways from the manuscript. It is hard to see the main findings as it is now.
Line 468: By blast score do you mean bit score? Or other? Was there a particular reason for the >60 value (usually 50 is significant)?
Author Response
#################Reviewer2
The manuscript entitled. “Hacking the Diversity of SARS-COV-2 and SARS-Like Coronaviruses in Human, Bat and Pangolin Populations,” focuses on the in-depth genetic analysis of SARS-CoV-2 genomes in various animals species. The analyses produced several notable findings including re-confirming the pangolin-CoV genomes cluster similarly with the Wuhan SARS-CoV-2 isolates, separation in host-associated CoVs in various ORFs specifically between pangolin-CoV and human SARS-COV-2, and identification of multiple variants that may have some important function. Overall, the information is interesting, well-written and important to further our understanding of molecular mechanisms associated and origins of SARS-CoV-2. Minor grammatical edits are suggested below:
Line 61: Correct analyse
##Thanks amended to "analysis"
Paragraph 73-76: Add a sentence or two about why we need to know and understand differing annotations, SNPs etc.
## Thank you for your comment. We have now updated the introduction (line 73-79).
Line 81-82: Are you using additional genomes that have not been utilized in previous studies?
## Whilst many of these genomes have been analysed in isolation, to our knowledge, there has not been a paper that collectively analysed all existing bat-CoV and pangolin-CoV.
By doing so, we have shown variations amongst the two groups.
We have now highlighted these in the discussion (line 222-226 and 262-264).
Line 84: Why did you target Germany in particular?
## The Charite SARS-CoV-2 genomes were included in this study as they provided high quality genome assemblies of isolates shortly after the original known outbreak date of late 2019.
Line 90: May want to add a sentence or two about previous research associated with RaTG13 and why it can be discounted.
## Thank you for your comment. We have now rewritten the section to avoid using the word, "discount". We have also added a reference noting RaTG13 being be closest known genome to SARS-CoV-2. (line 184)
Figure 1: The organization of the legend and figure makes it confusing. Please reorganize for clarity. Additionally,may want to put appendix A as in the main text of the manuscript. It is much easier to follow. As it is, it is difficult to differentiate the color changes in such a small section.
## Thank you for your suggestion. We have now replaced Figure 1 with the appendix A.
Finally, did you analyse the human genomes for codon usage or variants? If there were few variants in the pangolin/human genomes, why do you think that is? Did you compare the number of variants between species?
## We have used the SARS-CoV-2 (human coronavirus) for codon usage analysis. However, due to their high similarity to each other, the figure is less apparent as most SARS-CoV-2 have exactly the same codon usage as each other (therefore shown as a single purple dot in Figure 3). We have now made it more clear the number of SARS-CoV-2, pangolin-CoV and bat-CoV there is in each PCA plot in Figure 3 and its legend. As the variant calling in human isolated SARS-CoV-2 strains was not the focus of this study we have not performed this part for the Wuhan and Charite genomes. Other consortia work in much greater details are ongoing for such form of analysis e.g. COVID-19 Genomics UK (COG-UK) consortium \url{https://www.cogconsortium.uk/}
Line 168: Why was 18% used as a cutoff?
## We have tested a few thresholds and 18% allows a good trade off between the number of genes and bat-CoV genomes kept (most bat-CoV genes are so dissmilar to SARS-CoV-2 genes that they do not have a BLAST result, and some genes are missing due to gaps in the genomes). However, in response to Reviewer1’s comment, the multiple-gene PCA is now removed and replaced with the gene-by-gene PCA. The 18% filter is therefore no longer used in the analysis.
Line 173-175: Could the close clustering of these sequences suggest a similar origin? Or error in the assay?
## The clustering grouping the three bat-CoVs and pangolin-CoV is likely to reflect a similar origin rather than error in the assay, as these same four genomes are the only ones with BLAST results between their ORF8 and SARS-CoV-2 ORF8 (>85% similarity).
Line 233-234: From your data, can you suggest a more likely origin?
## Due to the main drive for objectivity of our approach.
The data shows a complex genomic origin shared between bats and pangolin coronavisues collected between 2017 and 2020. The under sampling and availability of the data has been pointed out in the discussion. We believe there are many missing links between the 3 hosts studied in this manuscript which makes aggregated suggestion of the virus origin very difficult and we didn't feel confident in doing so.
Discussion: Very long, please shorten. Highlight main take-aways from the manuscript. It is hard to see the main findings as it is now.
## Shortened and final paragraph rewritten to highlight main take-home messages
Line 468: By blast score do you mean bit score? Or other? Was there a particular reason for the >60 value (usually 50 is significant)?
## Yes it was bit score; we have updated the text. Both 50 and 60 are commonly used and we have selected the more stringent value for our analysis.
Round 2
Reviewer 1 Report
Thanks for all the changes and new analysis as per suggestions, paper looks good!!
Just one comment: Please maybe update ABSTRACT with some of the new results from new analysis.